# SABAL: Sparse Approximation-based Batch Active Learning

## Abstract

We propose a novel and general framework (*i.e.*, SABAL) that formulates batch active learning as a sparse approximation problem. SABAL aims to find a weighted subset from the unlabeled data pool such that the corresponding training loss function approximates its full data pool counterpart. We realize the general framework as a sparsity-constrained discontinuous optimization problem that explicitly balances uncertainty and representation for large-scale applications, for which we propose both greedy and iterative hard thresholding schemes. The proposed method can adapt to various settings, including both Bayesian and non-Bayesian neural networks. Numerical experiments show that that SABAL achieves state-of-the-art performance across different settings with lower computational complexity.

## 1 Introduction

Over the last decade, deep neural networks have achieved promising results in various learning tasks. However, obtaining labels for a complex training dataset can be challenging in practice, as the data annotation is usually a time-consuming process that may require professional knowledge in certain applications such as in medicine (Hoi et al., 2006; Shen et al., 2021). *Active Learning* (AL) (Settles, 2009) is commonly employed to mitigate the problem of scarce labeled data – enabling efficient model training with limited annotation costs. Given a partially labeled dataset, active learning ideally selects data samples that are the best for learning. Specifically, it aims to iteratively query the most helpful data to ask an oracle (human annotator) to annotate. The queried data samples can be added back to the labeled data pool, and the model is updated. This process is repeated until the model has achieved the desired performance. Intelligently identifying the most valuable data for annotation, also known as the query strategy, is the key problem in active learning.

A common strategy is to take the prediction uncertainty or data representation as the metric for data query. This *uncertainty-based* approach (Settles, 2009; Tong & Koller, 2001; Gal et al., 2017; Beluch et al., 2018) works by querying samples with high uncertainty, but often results in selecting correlated and redundant data samples in each batch (Kirsch et al., 2019; Ducoffe & Precioso, 2018). *Representation-based* approaches (Sener & Savarese, 2017; Yang & Loog, 2019) aim to select a subset of data that represents the whole unlabeled dataset, but tend to be computationally expensive and sensitive to batch sizes (Ash et al., 2019; Shui et al., 2020). More recently, several *hybrid* approaches that try to take both uncertainty and representation into consideration have shown advantages (Ash et al., 2019; Shui et al., 2020; Sinha et al., 2019). This paper takes this hybrid view towards an active learning framework that *balances the trade-off* between uncertainty and representation.

Besides hybrid approaches, deep Bayesian active learning has also gained attention due to recent advances in Bayesian deep learning. Several Bayesian approaches (Gal et al., 2017; Kirsch et al., 2019) leverage model uncertainty measurements (Gal & Ghahramani, 2015; 2016) determined by Bayesian neural networks, while other works (Pinsler et al., 2019) leverage progress in Bayesian Coreset problems (Zhang et al., 2021; Huggins et al., 2016; Campbell & Broderick, 2019). However, as most existing Bayesian approaches are explicitly designed for Bayesian neural networks, another goal of this paper is to propose a *general* method for both Bayesian and non-Bayesian models.

For deep models, it is reasonable to query a large batch of data simultaneously to reduce model update frequency. The batch selection approach is known as *batch active learning*. Taking an optimization perspective, finding the best batch is NP-hard in general. Two common approaches for such combinatorial problems are the greedy and clustering approaches. Greedy algorithms

select one data sample in sequence until the batch budget is exhausted (Kirsch et al., 2019; Bıyık et al., 2019; Chen & Krause, 2013). Here, specific conditions of the acquisition function such as submodularity (Nemhauser et al., 1978) are required to guarantee a good optimization result. Clustering algorithms regard cluster centers as their queried batch (Sener & Savarese, 2017; Ash et al., 2019), but can be computationally expensive. To our knowledge, except for Pinsler et al. (2019) that focus on the Bayesian models, so far active learning has rarely been studied from a *sparse approximation* perspective. This is despite the ubiquity of sparse approximation in signal processing for tasks such as dictionary learning (Aharon et al., 2006) and compressed sensing (Donoho, 2006) due to its performance for discovering a sparse representation while avoiding redundant information. Here we employ sparse approximation methods for batch active learning tasks.

Our main contributions are summarized in the following. We propose a novel and flexible Sparse Approximation-based Batch Active Learning framework, *i.e.*, SABAL. We show how SABAL generalizes batch active learning as a sparse approximation problem and can adapt to different settings and models. The central intuition of SABAL is finding a *weighted subset* from the unlabeled data pool so that its corresponding training loss approximates the *full-set* loss function in a function space. We realize the SABAL framework as an efficient finite-dimensional optimization problem: First, we derive an upper bound to balance the trade-off between uncertainty and representation in a principled way. Second, we approximate the loss functions using finite-dimensional approximation. This results in a sparsity-constrained discontinuous optimization problem, for which we propose several efficient optimization algorithms. We demonstrate the advantages of SABAL in experiments for both Bayesian and non-Bayesian batch active learning settings.

The structure of this manuscript is as follows. In Section 2, we formulate the general framework of SABAL, and in Section 3, we realize the framework into a finite-dimensional discontinuous sparse optimization problem. To solve the resulting optimization problem, we propose two optimization algorithms in Section 4. Related work are discussed in Section 5 and Appendix Section B. Results of our experiments are presented in section 6, and all proofs are provided in Appendix Section C.

## 2 BATCH ACTIVE LEARNING AS SPARSE APPROXIMATION

This section introduces the preliminaries and the general formulation of batch active learning as a sparse approximation problem.

**Preliminaries** Vectors are denoted as bold lower case letters, *e.g.*, $\boldsymbol{w} \in \mathbb{R}^n$. The $l_0$ pseudo-norm of a vector $\boldsymbol{w}$ is denoted as $\|\boldsymbol{w}\|_0$, *i.e.*, the number of non-zero elements of $\boldsymbol{w}$. We denote $\mathbb{R}_+ := [0, +\infty)$. Distributions are denoted in script, *e.g.*, $\mathscr{P}$, and a random variable is denoted by tilde, *e.g.*, $\tilde{\boldsymbol{y}} \sim \mathscr{P}$. We denote sets in calligraphy or in uppercase Greek alphabet (*e.g.*, $\mathcal{D}, \Theta$), and additionally we denote $[n] := \{1, 2, \dots, n\}$. In supervised learning, given a labeled training dataset $\mathcal{D}_l := \{(\boldsymbol{x}_i, \boldsymbol{y}_i)\}_{i=1}^{n_l}$, where we denote their domain to be $\boldsymbol{x} \in \mathcal{X}$ and $\boldsymbol{y} \in \mathcal{Y}$, the empirical goal is to minimize a loss function $L_l(\boldsymbol{\theta}) := \sum_{(\boldsymbol{x}_i, \boldsymbol{y}_i) \in \mathcal{D}_l} \ell(\boldsymbol{x}_i, \boldsymbol{y}_i; \boldsymbol{\theta})$ formed by the training dataset, where $\boldsymbol{\theta} \in \Theta \subset \mathbb{R}^m$ is the parameter of the model and $\ell$ is a loss function evaluated on individual pairs of data. Without loss of generality, we assume $\Theta \subset \mathbb{R}^m$ is compact and $\ell(\boldsymbol{x}, \boldsymbol{y}; \cdot) : \Theta \to \mathbb{R}$ is in a normed space $(\mathcal{L}(\Theta, \mathbb{R}), \|\cdot\|_\dagger)$ for all $\boldsymbol{x}, \boldsymbol{y}$. We further assume the constant function $f : \Theta \to 1$ is included in $\mathcal{L}(\Theta, \mathbb{R})$. The "$\dagger$" in the norm $\|\cdot\|_\dagger : \mathcal{L}(\Theta, \mathbb{R}) \to \mathbb{R}_+$, representing its definition is a placeholder that will be discussed later.

**Batch Active Learning** Besides the labeled dataset $\mathcal{D}_l$, there is an unlabeled dataset $\mathcal{D}_u := \{\boldsymbol{x}_j\}_{j=1}^{n_u}$ where the labels are unknown but could be acquired at a high cost through human labeling. Combining two datasets, the ideal loss function to minimize w.r.t. $\boldsymbol{\theta}$ is

$$\sum_{(\boldsymbol{x}_i, \boldsymbol{y}_i) \in \mathcal{D}_l} \ell(\boldsymbol{x}_i, \boldsymbol{y}_i; \boldsymbol{\theta}) + \sum_{\boldsymbol{x}_j \in \mathcal{D}_u} \ell(\boldsymbol{x}_j, \boldsymbol{y}_j^\star; \boldsymbol{\theta}), \tag{1}$$

where $\boldsymbol{y}_j^\star$ is the unknown true label corresponding to the data $\boldsymbol{x}_j$. Since acquiring true labels could be costly, we have to impose a budget $b$ $(b < n_u)$ on the number of label acquisitions. Therefore, the *batch active learning* problem is to find a subset $\mathcal{S} \subset \mathcal{D}_u$ such that we can obtain a good model by optimizing the following loss function w.r.t. $\boldsymbol{\theta}$,

$$\sum_{(\boldsymbol{x}_i, \boldsymbol{y}_i) \in \mathcal{D}_l} \ell(\boldsymbol{x}_i, \boldsymbol{y}_i; \boldsymbol{\theta}) + \sum_{\boldsymbol{x}_j \in \mathcal{S}} \ell(\boldsymbol{x}_j, \boldsymbol{y}_j^\star; \boldsymbol{\theta}), \quad \text{where } |\mathcal{S}| = b. \tag{2}$$

**Generalized Batch Active Learning**   We start our method by generalizing the classical formulation (equation 2) by considering an importance weight for each unlabeled data. That is, we aim to find a sparse non-negative vector $\boldsymbol{w} \in \mathbb{R}_+^{n_u}$ such that we can obtain a good model by optimizing the following loss function w.r.t. $\boldsymbol{\theta}$:

$$\sum_{(\boldsymbol{x}_i, \boldsymbol{y}_i) \in \mathcal{D}_l} \ell(\boldsymbol{x}_i, \boldsymbol{y}_i; \boldsymbol{\theta}) + \sum_{\boldsymbol{x}_j \in \mathcal{D}_u} w_j \ell(\boldsymbol{x}_j, \boldsymbol{y}_j^\star; \boldsymbol{\theta}), \quad \text{where } \|\boldsymbol{w}\|_0 = b. \tag{3}$$

A *key question* now is—what is the criterion for a good $\boldsymbol{w}$? Comparing the ideal loss function (equation 1) and the sparse importance weighted loss (equation 3), the only difference is their unlabeled data loss functions. Therefore, a straight-forward informal criterion for a good importance weight $\boldsymbol{w}$ is that the two unlabeled data loss functions are close to each other, *i.e.*,

$$L_{\boldsymbol{w}}^\star(\boldsymbol{\theta}) := \frac{1}{b} \sum_{\boldsymbol{x}_j \in \mathcal{D}_u} w_j \ell(\boldsymbol{x}_j, \boldsymbol{y}_j^\star; \boldsymbol{\theta}) \quad \approx \quad L^\star(\boldsymbol{\theta}) := \frac{1}{n_u} \sum_{\boldsymbol{x}_j \in \mathcal{D}_u} \ell(\boldsymbol{x}_j, \boldsymbol{y}_j^\star; \boldsymbol{\theta}).$$

However, as the true labels are unknown, we cannot compute $L_{\boldsymbol{w}}^\star$ and $L^\star$. Luckily, we can have an estimator for the true labels, *i.e.*, estimation based on the labeled data $p(\tilde{\boldsymbol{y}}_j \mid \boldsymbol{x}_j, \mathcal{D}_l)$ or an approximation of it. Denote $\mathscr{P}(\boldsymbol{x}_j)$ as an estimated distribution, so $\tilde{\boldsymbol{y}}_j \sim \mathscr{P}(\boldsymbol{x}_j)$, then the informal criterion for a good importance weight $\boldsymbol{w}$ then becomes

$$\tilde{L}_{\boldsymbol{w}}(\boldsymbol{\theta}) := \frac{1}{b} \sum_{\boldsymbol{x}_j \in \mathcal{D}_u} w_j \ell(\boldsymbol{x}_j, \tilde{\boldsymbol{y}}_j; \boldsymbol{\theta}) \quad \approx \quad \tilde{L}(\boldsymbol{\theta}) := \frac{1}{n_u} \sum_{\boldsymbol{x}_j \in \mathcal{D}_u} \ell(\boldsymbol{x}_j, \tilde{\boldsymbol{y}}_j; \boldsymbol{\theta}). \tag{4}$$

Thus, we are one step closer to evaluating the quality of a weighted selection. The next question is how to measure the difference between $\tilde{L}$ and $\tilde{L}_{\boldsymbol{w}}$.

**Difference Between Two Loss Functions**   Given the two loss functions $\tilde{L}, \tilde{L}_{\boldsymbol{w}} \in \mathcal{L}(\Theta, \mathbb{R})$, where $\mathcal{L}(\Theta, \mathbb{R})$ is equipped with the norm $\|\cdot\|_\dagger$, a straight-forward measurement of the difference between them is $\|\tilde{L} - \tilde{L}_{\boldsymbol{w}}\|_\dagger$. However, observing that the optimization of a loss function is shift-invariant, the difference between two loss functions should also be shift-invariant. For example, for $\forall L \in \mathcal{L}(\Theta, \mathbb{R})$ we have $\arg\min_{\boldsymbol{\theta} \in \Theta}(L(\boldsymbol{\theta}) + c) = \arg\min_{\boldsymbol{\theta} \in \Theta} L(\boldsymbol{\theta})$ for $\forall c \in \mathbb{R}$, implying that $L + c$ should be treated the same as $L$. Therefore, to account for the shift-invariance, we define $q : \mathcal{L}(\Theta, \mathbb{R}) \to \mathbb{R}_+$ as

$$q(L) := \inf_{c \in \mathbb{R}} \|L + c\|_\dagger, \qquad \forall L \in \mathcal{L}(\Theta, \mathbb{R}). \tag{5}$$

Note that *we abuse the notation* a bit, *i.e.*, the $c$ in $L + c$ should be the constant function that maps every $\boldsymbol{\theta} \in \Theta$ to $c$. The above definition has some nice properties that make it a good difference measurement of two loss functions, as proved in proposition C.1 in the appendix. In particular, $q(\cdot)$ satisfies *the triangle inequality*, and $q(L + c) = q(L)$ for any constant $c$. Therefore, we can formulate the generalized batch active learning problem as the following sparse approximation problem.

**Problem 1** (Sparse Approximation-based Batch Active Learning). *Given the shift-invariant seminorm $q$ induced by the norm $\|\cdot\|_\dagger$ (equation 5), and a label estimation distribution $\mathscr{P}$, the generalized batch active learning problem (equation 4) is formally defined as*

$$\underset{\boldsymbol{w} \in \mathbb{R}_+^{n_u}}{\arg\min} \quad \mathbb{E}_{\mathscr{P}}[q(\tilde{L} - \tilde{L}_{\boldsymbol{w}})] \qquad s.t. \qquad \|\boldsymbol{w}\|_0 = b, \tag{6}$$

*where $\mathbb{E}_{\mathscr{P}}$ stands for the expectation over $\tilde{\boldsymbol{y}}_j \sim \mathscr{P}(\boldsymbol{x}_j)$ for $\forall j \in [n_u]$.*

Problem 1 (SABAL) offers a general framework for batch active learning and can be applied with various settings, *i.e.*, both the norm $\|\cdot\|_\dagger$ and the individual loss function $\ell$ can be chosen based on specific problems and applications. In the next section, we introduce two practical realizations of equation 6 for Bayesian and non-Bayesian active learning respectively.

## 3   SPARSE APPROXIMATION AS FINITE-DIMENSIONAL OPTIMIZATION

In this section, we transform the sparse approximation problem (equation 6) into a finite-dimensional sparse optimization problem. First, we address an issue regarding the sampling of $\mathbb{E}_{\mathscr{P}}$. Then, we discuss some concrete choices of $\mathscr{P}$ and $\|\cdot\|_\dagger$ that lead to a finite-dimensional sparse optimization.

**Addressing the Sampling Issue** In equation 6, the expectation $\mathbb{E}_{\mathscr{P}}$ is taken over the product space of $(\tilde{\boldsymbol{y}}_1, \ldots, \tilde{\boldsymbol{y}}_{n_u})$ and each sample has to be remembered for future optimization, which can be intractable for large datasets. However, it has an upper bound where the complexity of the optimization is independent of the number of samples from $\mathscr{P}$. First, by the triangle inequality

$$\mathbb{E}_{\mathscr{P}}[q(\tilde{L} - \tilde{L}_{\boldsymbol{w}})] = \mathbb{E}_{\mathscr{P}}[q(\tilde{L} - \mathbb{E}_{\mathscr{P}}[\tilde{L}] + \mathbb{E}_{\mathscr{P}}[\tilde{L}] - \mathbb{E}_{\mathscr{P}}[\tilde{L}_{\boldsymbol{w}}] + \mathbb{E}_{\mathscr{P}}[\tilde{L}_{\boldsymbol{w}}] - \tilde{L}_{\boldsymbol{w}})]$$

$$\leq \underbrace{\mathbb{E}_{\mathscr{P}}[q(\tilde{L} - \mathbb{E}_{\mathscr{P}}[\tilde{L}])]}_{(i):\ \text{variance}} + \mathbb{E}_{\mathscr{P}}[q(\tilde{L}_{\boldsymbol{w}} - \mathbb{E}_{\mathscr{P}}[\tilde{L}_{\boldsymbol{w}}])] + \underbrace{q(\mathbb{E}_{\mathscr{P}}[\tilde{L}] - \mathbb{E}_{\mathscr{P}}[\tilde{L}_{\boldsymbol{w}}])}_{(ii):\ \text{approximation bias}}. \quad (7)$$

We can see that it offers a trade-off between bias and variance, where the bias term is immediately tractable by expanding $\tilde{L}, \tilde{L}_{\boldsymbol{w}}$:

$$(ii) = q(\mathbb{E}_{\mathscr{P}}[\tfrac{1}{n_u} \textstyle\sum_{\boldsymbol{x}_j \in \mathcal{D}_u} \ell(\boldsymbol{x}_j, \tilde{\boldsymbol{y}}_j; \cdot)] - \mathbb{E}_{\mathscr{P}}[\tfrac{1}{b} \textstyle\sum_{\boldsymbol{x}_j \in \mathcal{D}_u} w_j \ell(\boldsymbol{x}_j, \tilde{\boldsymbol{y}}_j; \cdot)])$$

$$= q((\tfrac{1}{n_u} \textstyle\sum_{\boldsymbol{x}_j \in \mathcal{D}_u} \mathbb{E}_{\mathscr{P}(\boldsymbol{x}_j)}[\ell(\boldsymbol{x}_j, \tilde{\boldsymbol{y}}_j; \cdot)]) - (\tfrac{1}{b} \textstyle\sum_{\boldsymbol{x}_j \in \mathcal{D}_u} w_j \mathbb{E}_{\mathscr{P}(\boldsymbol{x}_j)}[\ell(\boldsymbol{x}_j, \tilde{\boldsymbol{y}}_j; \cdot)])). \quad (8)$$

It remains to address the variance term *(i)*. Recall that the more accurate $\mathscr{P}$ is, the more accurate our approximation is. Given the decision $w_j > 0$, if the label of $\boldsymbol{x}_j$ is acquired, *i.e.*, the oracle (human annotator) will offer us its true label $\boldsymbol{y}_j^\star$, and the labeling distribution would be improved. That being said, the distribution of $\tilde{\boldsymbol{y}}_j$ given $\boldsymbol{x}_j$ and $w_j > 0$ will be concentrated on its true label $\boldsymbol{y}_j^\star$, *i.e.*,

$$\tilde{\boldsymbol{y}}_j \sim \mathscr{P}_{\boldsymbol{w}}(\boldsymbol{x}_j) := \begin{cases} \mathscr{P}(\boldsymbol{x}_j) & \text{if } w_j = 0 \\ \delta_{\boldsymbol{y}_j^\star} & \text{if } w_j > 0 \end{cases}, \qquad \text{where } \boldsymbol{w} \in \mathbb{R}_+^{n_u} \quad (9)$$

where $\delta_{\boldsymbol{y}^\star}$ denotes the distribution that $\tilde{\boldsymbol{y}}_j$ can only be $\boldsymbol{y}_j^\star$. However, the improved distribution $\mathscr{P}_{\boldsymbol{w}}$ is not known before the acquisition of the true labels $\boldsymbol{y}_j^\star$ for $w_j > 0$. Fortunately, although $\mathscr{P}_{\boldsymbol{w}}(\boldsymbol{x}_j)$ is not known, it is known that the corresponding variance for $\tilde{\boldsymbol{y}}_j$ would be zero no matter what its label is. Applying this trick, we show in the following proposition that the term *(i)* with the improved label distribution $\mathscr{P}_{\boldsymbol{w}}$ has an upper bound that does not require to know the true labels.

**Proposition 3.1.** *Let $\boldsymbol{w} \in \mathbb{R}_+^{n_u}$ and $\|\boldsymbol{w}\|_0 = b$, by replacing the $\mathscr{P}$ by the improved estimation distribution $\mathscr{P}_{\boldsymbol{w}}$ (equation 9) into (i) in equation 7, we have*

$$\mathbb{E}_{\mathscr{P}_{\boldsymbol{w}}}[q(\tilde{L} - \mathbb{E}_{\mathscr{P}_{\boldsymbol{w}}}[\tilde{L}])] + \mathbb{E}_{\mathscr{P}_{\boldsymbol{w}}}[q(\tilde{L}_{\boldsymbol{w}} - \mathbb{E}_{\mathscr{P}_{\boldsymbol{w}}}[\tilde{L}_{\boldsymbol{w}}])] \leq \sum_{\boldsymbol{x}_j \in \mathcal{D}_u} \mathbf{1}(w_j = 0) \cdot \sigma_j, \quad (10)$$

*where $\sigma_j := \frac{1}{n_u} \mathbb{E}_{\mathscr{P}(\boldsymbol{x}_j)}[q(\ell(\boldsymbol{x}_j, \tilde{\boldsymbol{y}}_j; \cdot) - \mathbb{E}_{\mathscr{P}(\boldsymbol{x}_j)}[\ell(\boldsymbol{x}_j, \tilde{\boldsymbol{y}}_j; \cdot)])]$ is the individual variance, and $\mathbf{1}(\cdot)$ is the indicator function.*

Therefore, combining equation 8 and equation 10, we have a more tractable form of the sparse approximation, *i.e.*,

$$\operatorname*{arg\,min}_{\boldsymbol{w} \in \mathbb{R}_+^{n_u}} \quad q(\mathbb{E}_{\mathscr{P}}[\tilde{L}] - \mathbb{E}_{\mathscr{P}}[\tilde{L}_{\boldsymbol{w}}]) + \sum_{\boldsymbol{x}_j \in \mathcal{D}_u} \mathbf{1}(w_j = 0) \cdot \sigma_j \qquad \text{s.t.} \qquad \|\boldsymbol{w}\|_0 = b, \quad (11)$$

Intuitively, such decomposition of *bias* and *variance* naturally provides metrics of *uncertainty* and *representation* for active learning, where the variance itself is a metric of uncertainty, meanwhile the bias term measures how well a subset of selected data can represent the whole unlabeled data. Now, it remains to specify the choice of $\| \cdot \|_\dagger$, *i.e.*, the norm that induces $q$ (equation 5).

**Formulation of the Finite-Dimensional Optimization** We consider two concrete choices of the $\| \cdot \|_\dagger$ for Bayesian and non-Bayesian settings respectively.

1. In the Bayesian setting, we can easily sample $\boldsymbol{\theta}_i \sim \pi := p(\boldsymbol{\theta} \mid \mathcal{D}_l)$ from the posterior. Utilizing the posterior, we make the norm $\| \cdot \|_\dagger$ more concrete by considering the $L^2(\pi)$-norm, *i.e.*, $\|L\|_\pi^2 = \mathbb{E}_{\boldsymbol{\theta} \sim \pi}[L(\boldsymbol{\theta})^2]$. Accordingly,

   $$q(L)^2 = \inf_{c \in \mathbb{R}} \|L + c\|_\pi^2 = \inf_{c \in \mathbb{R}} \mathbb{E}_{\boldsymbol{\theta} \sim \pi}[(L(\boldsymbol{\theta}) + c)^2] = \mathbb{E}_{\boldsymbol{\theta} \sim \pi}[(L(\boldsymbol{\theta}) - \mathbb{E}_{\boldsymbol{\theta} \sim \pi}[L(\boldsymbol{\theta})])^2]. \quad (12)$$

   The posterior $\pi$ tells us where and how to evaluate the "magnitude" of $L$. Noting that equation 12 is in the form of an expectation, we can draw $m$ samples $\boldsymbol{\theta}_i \sim \pi$ to approximate it. Denote $\boldsymbol{g} := \frac{1}{\sqrt{m}}[\ldots, (L(\boldsymbol{\theta}_i) - \bar{L}), \ldots]_{i=1\ldots m}^\top \in \mathbb{R}^m$ where $\bar{L} := \frac{1}{m} \sum_{i=1}^m L(\boldsymbol{\theta}_i)$. equation 12 becomes

   $$q(L)^2 \approx \frac{1}{m} \sum_{i=1}^m (L(\boldsymbol{\theta}_i) - \bar{L})^2 = \|\boldsymbol{g}\|_2^2, \quad (13)$$

   where $\|\boldsymbol{g}\|_2$ is simply the Euclidean norm of the $m$-dimensional vector $\boldsymbol{g}$.

2. In the non-Bayesian setting, we evaluate the loss function in a local "window" based on the current model. We consider the $\|\cdot\|_\infty$-norm over a Euclidean ball $\mathcal{B}_r(\boldsymbol{\theta}_0)$ of radius $r$ centered at the current model parameter $\boldsymbol{\theta}_0$, i.e., $\|L\|_\infty = \max_{\boldsymbol{\theta} \in \mathcal{B}_r(\boldsymbol{\theta}_0)} |L(\boldsymbol{\theta})|$. Moreover, in the Euclidean ball we approximate $L(\boldsymbol{\theta}) \approx L(\boldsymbol{\theta}_0) + \nabla L(\boldsymbol{\theta}_0)^\top (\boldsymbol{\theta} - \boldsymbol{\theta}_0)$. Therefore, we have

$$q(L) = \inf_{c \in \mathbb{R}} \|L + c\|_\infty = \inf_{c \in \mathbb{R}} \max_{\boldsymbol{\theta} \in \mathcal{B}_r(\boldsymbol{\theta}_0)} |L(\boldsymbol{\theta}) + c|$$

$$\approx \inf_{c \in \mathbb{R}} \max_{\boldsymbol{\theta} \in \mathcal{B}_r(\boldsymbol{\theta}_0)} |L(\boldsymbol{\theta}_0) + \nabla L(\boldsymbol{\theta}_0)^\top (\boldsymbol{\theta} - \boldsymbol{\theta}_0) + c| = r\|\nabla L(\boldsymbol{\theta}_0)\|_2. \tag{14}$$

Note that $\|\nabla L(\boldsymbol{\theta}_0)\|_2$ is the Euclidean norm of the gradient vector $\nabla L(\boldsymbol{\theta}_0) \in \mathbb{R}^m$.

To estimate the label distribution, $\mathscr{P}(\boldsymbol{x}_j) = p(\tilde{y}_j \mid \boldsymbol{x}_j, \mathcal{D}_l)$ can be directly applied on Bayesian models by estimating the predictive model posterior. For non-Bayesian models, one could utilize the calibrated model prediction (Guo et al., 2017) as the label distribution. Finally, plugging either of the two approximations of $q(L)$ into equation 11, and squaring all of the terms for the ease of optimization, we can formulate the sparse approximation problem as the following finite-dimensional optimization problem, where $\alpha > 0$ offers a trade-off between bias and variance.

$$\arg\min_{\boldsymbol{w} \in \mathbb{R}_+^{n_u}} \|\boldsymbol{v} - \Phi\boldsymbol{w}\|_2^2 + \alpha \sum_{\boldsymbol{x}_j \in \mathcal{D}_u} \mathbf{1}(w_j = 0) \cdot \sigma_j^2 \quad \text{s.t.} \quad \|\boldsymbol{w}\|_0 = b, \tag{15}$$

where we denote $\boldsymbol{v} \in \mathbb{R}^m$, $\Phi \in \mathbb{R}^{m \times n_u}$ and $\sigma_j$ as

$$\boldsymbol{v} := \frac{1}{n_u} \sum_{j=1}^{n_u} \mathbb{E}_{\mathscr{P}(\boldsymbol{x}_j)}[\boldsymbol{g}_j(\tilde{\boldsymbol{y}}_j)], \qquad \Phi := \frac{1}{b}(\mathbb{E}_{\mathscr{P}(\boldsymbol{x}_1)}[\boldsymbol{g}_1(\tilde{\boldsymbol{y}}_1)], \ldots, \mathbb{E}_{\mathscr{P}(\boldsymbol{x}_{n_u})}[\boldsymbol{g}_{n_u}(\tilde{\boldsymbol{y}}_{n_u})]),$$

$$\sigma_j = \frac{1}{n_u} \mathbb{E}_{\mathscr{P}(\boldsymbol{x}_j)}[\|\boldsymbol{g}_j(\tilde{\boldsymbol{y}}_j) - \mathbb{E}_{\mathscr{P}(\boldsymbol{x}_j)}[\boldsymbol{g}_j(\tilde{\boldsymbol{y}}_j)]\|_2],$$

$$\boldsymbol{g}_j(\tilde{\boldsymbol{y}}_j) := \begin{cases} [\ldots, (\ell(\boldsymbol{x}_j, \tilde{\boldsymbol{y}}_j; \boldsymbol{\theta}_i) - \bar{\ell}), \ldots]_{i=1\ldots m}^\top, & \bar{\ell} := \frac{1}{m} \sum_{i=1}^m \ell(\boldsymbol{x}_j, \tilde{\boldsymbol{y}}_j; \boldsymbol{\theta}_i) \quad \text{if use (13)} \\ \nabla \ell(\boldsymbol{x}_j, \tilde{\boldsymbol{y}}_j; \boldsymbol{\theta}_0) & \text{if use (14)}. \end{cases} \tag{16}$$

In practice, it is often the case that the number of parameters is less than the number of samples, i.e., $m < n_u$, even for over-parameterized neural networks where the gradient of the last layer is commonly used to represent the full-model gradient (Katharopoulos & Fleuret, 2018; Ash et al., 2019). Therefore, if the batch size is big, i.e., $b > m$, the approximation bias $\|\boldsymbol{v} - \Phi\boldsymbol{w}\|_2^2$ may be under-determined with infinitely many $\boldsymbol{w}$ to make $\boldsymbol{v} = \Phi\boldsymbol{w}$, and the optimization (equation 15) may be "overfitted". To make our method more stable, we include a $\ell_2$ regularizer $\beta\|\boldsymbol{w} - \mathbf{1}\|_2^2$ with $\beta > 0$. Finally, since $w_j \geq 0$, minimizing $\alpha \sum_{\boldsymbol{x}_j \in \mathcal{D}_u} \mathbf{1}(w_j = 0) \cdot \sigma_j^2$ is equivalent to minimizing $-\alpha \sum_{\boldsymbol{x}_j \in \mathcal{D}_u} \mathbf{1}(w_j > 0) \cdot \sigma_j^2$. Consequently, we have the following optimization problem.

**Problem 2** (Sparse Approximation as Finite-dimensional Optimization). *The finite-dimensional optimization for generalized batch active learning is*

$$\arg\min_{\boldsymbol{w} \in \mathbb{R}_+^{n_u}} \|\boldsymbol{v} - \Phi\boldsymbol{w}\|_2^2 - \alpha \sum_{\boldsymbol{x}_j \in \mathcal{D}_u} \mathbf{1}(w_j > 0) \cdot \sigma_j^2 + \beta\|\boldsymbol{w} - \mathbf{1}\|_2^2 \quad s.t. \quad \|\boldsymbol{w}\|_0 = b. \tag{17}$$

While simplified, the result is a sparse discontinuous optimization problem generally difficult to solve. In the next section, we propose two optimization algorithms for equation 17 by exploiting its unique properties. The overall procedure of SABAL in practice is presented in Algorithm 3 Appendix A.

## 4 OPTIMIZATION ALGORITHMS

This section focuses on optimizing Problem 2. Rewrite equation 17 $f(\boldsymbol{w}) := f_1(\boldsymbol{w}) + f_2(\boldsymbol{w})$, where

$$f_1(\boldsymbol{w}) := \|\boldsymbol{v} - \Phi\boldsymbol{w}\|_2^2 + \beta\|\boldsymbol{w} - \mathbf{1}\|_2^2, \qquad f_2(\boldsymbol{w}) := -\alpha \sum_{\boldsymbol{x}_j \in \mathcal{D}_u} \mathbf{1}(w_j > 0) \cdot \sigma_j^2.$$

The optimization has two major difficulties, i.e., the nonconvex sparsity constraint $\|\boldsymbol{w}\|_0 = b$ and the discontinuous objective function $f_2$. When it comes to sparsity-constrained optimization, there are

two schemes that are widely considered — greedy (Nemhauser et al., 1978; Campbell & Broderick, 2019) and iterative hard thresholding (IHT) (Zhang et al., 2021; Khanna & Kyrillidis, 2018). However, Problem 2 introduces the new difficulty other than the sparsity constraint, *i.e.*, the discontinuous component $f_2$, which violate the assumptions of many of these methods which require the use of gradient. Instead, we propose two algorithms (Algorithm 1&2) specifically for Problem 2 under the two schemes respectively, while incorporating the discontinuity.

We introduce some notations used in this section. Given a vector $\boldsymbol{g}$, we denote $[\boldsymbol{g}]_+$ as $\boldsymbol{g}$ with its negative elements set to 0. For an index $j$, we denote $g_j$ or $(\boldsymbol{g})_j$ to be its $j^{th}$ element. For an index set $\mathcal{S}$, we denote $[\boldsymbol{g}]_{\mathcal{S}}$ to be the vector where $([\boldsymbol{g}]_{\mathcal{S}})_j = (\boldsymbol{g})_j$ if $j \in \mathcal{S}$ and $([\boldsymbol{g}]_{\mathcal{S}})_j = 0$ if $j \notin \mathcal{S}$. Moreover, we denote $\boldsymbol{e}^j$ to be the unit vector where $(\boldsymbol{e}^j)_j = 1$ and $(\boldsymbol{e}^j)_i = 0$ for $\forall i \neq j$.

Although two algorithms use different schemes, they share the same two sub-procedures: a line search and de-bias step (Algorithm 4 and 5 in Appendix D), which significantly improve the optimization performance (Zhang et al., 2021). The line search sub-procedure optimally solves the problem $\arg\min_{\mu \in \mathbb{R}} f_1(\boldsymbol{w} - \mu\boldsymbol{u})$, *i.e.*, given a direction $\boldsymbol{u}$, what is the best step size $\mu$ to move the $\boldsymbol{w}$ along $\boldsymbol{u}$. The de-bias sub-procedure adjusts a sparse $\boldsymbol{w}$ in its own sparse support for a better solution.

**Opt. Algorithm: Greedy** The core idea of the greedy approach is noted in line 3 Algorithm 1, where it chooses an index $j$ to move a step of size $\tau$ that minimizes the objective, *i.e.*, $j \leftarrow \arg\min_{j \in [n_u]\setminus\mathcal{S}} (f_1(\boldsymbol{w} + \tau\boldsymbol{e}^j) - f_1(\boldsymbol{w})) + (f_2(\boldsymbol{w} + \tau\boldsymbol{e}^j) - f_2(\boldsymbol{w}))$. By approximating $f_1(\boldsymbol{w} + \tau\boldsymbol{e}^j) - f_1(\boldsymbol{w})$ by its first-order approximation $\langle \nabla f_1(\boldsymbol{w}), \tau\boldsymbol{e}^j \rangle$, and noting that $f_2(\boldsymbol{w} + \tau\boldsymbol{e}^j) - f_2(\boldsymbol{w}) = -\alpha\sigma_j^2$, we have the greedy step (line 3) in Algorithm 1. After choosing the index $j$ to include, line 5 chooses an optimal step to move, followed by a de-bias step that further improves the solution in the current sparse support $\text{supp}(\boldsymbol{w})$.

**Opt. Algorithm: Proximal iterative hard thresholding** The core idea of the proximal IHT (Algorithm 2) is noted in line 6, where it combines both the hard thresholding and the proximal operator. It minimizes the discontinuous $f_2$ in a neighbourhood of the solution $\boldsymbol{s}$ obtained by minimizing $f_1$, while satisfying the constraints. As discussed in the section D, the inner optimization (line 6) can be done optimally by simply picking the top-$b$ elements from $n_u$ elements. After this core step, a de-bias step improves the solution $\boldsymbol{w}$ within its sparse support, followed by a momentum step.

---

**Algorithm 1:** SABAL-Greedy

**Parameter:** sparsity $b$; step size $\tau$.

1   $\boldsymbol{w} \leftarrow \boldsymbol{0}$;   $\mathcal{S} \leftarrow \emptyset$
2   **repeat**
3     $j \leftarrow \underset{j \in [n_u]\setminus\mathcal{S}}{\arg\min} \ \tau(\nabla f_1(\boldsymbol{w}))_j - \alpha\sigma_j^2$
4     $\mathcal{S} \leftarrow \mathcal{S} \cup \{j\}$       *(update selection)*
5     $\mu \leftarrow \text{LineSearch}(\boldsymbol{e}^j, \boldsymbol{w})$
6     $\boldsymbol{w} \leftarrow \text{De-bias}(\boldsymbol{w} - \mu\boldsymbol{e}^j)$
7     $w_j \leftarrow 0$ for $\forall w_j < 0$     $(\boldsymbol{w} \in \mathbb{R}_+^{n_u})$
8   **until** $|\mathcal{S}| = b$;

**Return:** $\boldsymbol{w}$

---

**Algorithm 2:** SABAL-IHT

**Parameter:** sparsity $b$; number of iterations $T$.

1   $\boldsymbol{w} \leftarrow \boldsymbol{0}$;   $\boldsymbol{z} \leftarrow \boldsymbol{0}$
2   **repeat**
3     $\boldsymbol{w}' \leftarrow \boldsymbol{w}$       *(save previous $\boldsymbol{w}$)*
4     $\mu \leftarrow \text{LineSearch}(\nabla f_1(\boldsymbol{z}), \boldsymbol{z})$
5     $\boldsymbol{s} \leftarrow \boldsymbol{z} - \mu\nabla f_1(\boldsymbol{z})$    *(gradient descent)*
6     $\boldsymbol{w} \leftarrow \underset{\boldsymbol{w} \in \mathbb{R}_+^{n_u}, \|\boldsymbol{w}\|_0 \leq b}{\arg\min} \ \frac{1}{2}\|\boldsymbol{w} - \boldsymbol{s}\|_2^2 + f_2(\boldsymbol{w})$
7     $\boldsymbol{w} \leftarrow \text{De-bias}(\boldsymbol{w})$
8     $w_j \leftarrow 0$ for $\forall w_j < 0$     $(\boldsymbol{w} \in \mathbb{R}_+^{n_u})$
9     $\tau \leftarrow \text{LineSearch}(\boldsymbol{w} - \boldsymbol{w}', \boldsymbol{w})$
10    $\boldsymbol{z} \leftarrow \boldsymbol{w} - \tau(\boldsymbol{w} - \boldsymbol{w}')$    *(momentum)*
11   **until** $T$ *iterations*;

**Return:** $\boldsymbol{w}$

---

**Complexity Analysis** We analyze the time complexity of the proposed algorithms with respect to the number of data samples $n$, and the batch size $b$ of batch active learning. Except for line 6 Algorithm 2, all steps are of time complexity $O(n)$. The line 6 Algorithm 2 is finding the $b$ smallest elements, which can be done in $O(n\log(b))$. Therefore, the time complexity for SABAL-Greedy is $O(nb)$, and the time complexity for SABAL-IHT is $O(n\log(b))$. Comparing to the time complexity $O(nb^2)$ of the state-of-the-art method BADGE (Ash et al., 2019), the two proposed algorithms can be much faster, especially with a large batch size $b$ used in practice.

## 5   RELATED WORK

Our method has several characteristics: (1) it's a hybrid active learning approach. (2) it's a general framework with building blocks easily adapted to both Bayesian and non-Bayesian settings. (3) it

formulates data acquisition as a sparse approximation problem. This section focuses on discussing some most relevant works, and explain how they motivate and compare to our work. Other related works are discussed in Appendix B.

As data acquisitions with the trade-off between uncertainty and representation have attracted attention, several recent works have proposed hybrid active learning methods. One of the state-of-art methods, BADGE (Ash et al., 2019), captures uncertainty through the lens of gradients, and samples diverse batches on the gradient embedding by the $k$-MEANS++ seeding algorithm. However, one of the downsides of BADGE is the high run-time complexity, as data acquisition speed is crucial in practice. Sinha et al. (2019) train a Variational Autoencoder and a discriminator in an adversarial fashion. The discriminator predicts a sample as unlabeled based on its likelihood of representativeness, and a batch of samples with the lowest confidence will be queried, but their adversarial method is difficult to apply to general and Bayesian neural networks. Our proposed method explicitly balances the trade-offs between uncertainty and representation by bias and variance decomposition.

Coreset selection is a common high-level idea used in active learning, and methods vary in how one characterizes the closeness of a chosen coreset to the full-set. Sener & Savarese (2017) characterizes the closeness as how much a coreset covers the full-set in the Euclidean distance in a feature space. It derives an upper bound for the coreset loss based on the Lipschitz continuity and transforms the original problem to a KCenter problem. However, their method relies on good feature representation, which is not always guaranteed in practice. Pinsler et al. (2019) is mainly based on existing Bayesian inference literature, especially the Bayesian Coreset problem (Campbell & Broderick, 2019). They characterizes the closeness as how much the core-set log-posterior approximates the full-set log-posterior, with the log-posterior directly derived from the Bayes' rule. However, their problem formulation relies on the Bayesian setting and Bayesian models, and conducting posterior inference is non-trivial for non-Bayesian models. In constrast, we propose SABAL, which characterizes the closeness in a more general sense, i.e., through a semi-norm function directly on the difference between the coreset loss function and the full-set loss function.

## 6 EXPERIMENT RESULTS

We demonstrate that SABAL is a flexible batch active learning framework with relatively small time complexity by evaluating its performance on image classification tasks with various models under different settings. First, using Bayesian neural networks, we show the effectiveness of SABAL on Bayesian batch active learning. Next, we demonstrate that SABAL can also adapt well to general batch active learning with general neural networks. Finally, we show that SABAL also has runtime advantages compared to other state-of-art methods. We also conduct an ablation study to show how SABAL balances the trade-offs between uncertainty and representation in Appendix E.1.

We have a fixed training, validation, and testing set in each experiment. The model is initially trained on small amounts of labeled data randomly selected from the training set and then iteratively performs the data acquisition and annotation. The model is reinitialized and retrained at the beginning of each active learning iteration. After the model is well trained, its testing accuracy is evaluated on the testing set as a measure of the performance. All experiments are repeated multiple times using 5 random seeds (3 for the small model LeNet-5 (LeCun et al., 2015)), and the results are reported as mean and standard deviations. The performance of each iteration are shown in learning curve plots. To better visualize the overall performance of AL methods, We also measure the area under curve (AUC) scores of the learning curve of different AL methods across different datasets. The top two AUC scores are highlighted in **bald**. We implement SABAL using both proximal IHT and greedy as two different optimization methods for the sparse approximation, denoted as *SABAL-IHT* and *SABAL-Greedy*, compared with following baselines in literature: **(1) Random**: A naive baseline that selects a batch of data uniformly at random. **(2) BALD** (Houlsby et al., 2011): An uncertainty-based Bayesian method that selects a batch of data with maximum mutual information between model parameters and predictions. **(3) Entropy** (Wang & Shang, 2014): An uncertainty-based non-Bayesian method that selects a batch of data with maximum entropy of the model predictions $\mathbb{H}(y_i \mid x_i; \theta)$. **(4) KCenter** (Sener & Savarese, 2017): A representation-based non-Bayesian method that reformulates the coreset selection as a KCenter problem in the feature embedding space. **(5) BADGE** (Batch Active Learning by Diverse Gradient Embeddings) (Ash et al., 2019): A hybrid non-Bayesian method that samples a diverse batch of data using the $k$-MEANS++ seeding algorithm.

**(6) Bayesian Coreset** (Pinsler et al., 2019): A Bayesian batch active learning approach based on the Bayesian Coreset problem (Huggins et al., 2016; Campbell & Broderick, 2019).

**SABAL for Bayesian Active Learning** We first implement our experiments on Bayesian neural networks and perform Bayesian active learning on Fashion MNIST (LeCun et al., 1998), CIFAR-10 (Krizhevsky et al., 2009), and CIFAR-100 (Krizhevsky et al., 2009). For fair comparison, we keep the same experiment settings of Pinsler et al. (2019), using a Bayesian neural network consisting of a ResNet-18 (He et al., 2016) feature extractor. The posterior inference is obtained by variational inference (Wainwright & Jordan, 2008; Blundell et al., 2015) at the last layer, and model predictive posteriors $p(\tilde{\boldsymbol{y}}_j \mid \boldsymbol{x}_j, \mathcal{D}_l)$ are estimated using 100 samples. Equation 13 is used to solve the finite-dimensional optimization problem, because sampling from the posterior distribution in a Bayesian neural network will be efficient by leveraging the local reparameterization trick (Kingma et al., 2015). Besides Random, we mainly focus on comparing with state-of-the-art approaches specifically designed for Bayesian active learning: BALD and Bayesian Coreset.

We then evaluate SABAL's performace. On Fashion MNIST dataset, we use 100 samples for random projections, 1000 seed data, and query 1000 samples for 9 iterations. On CIFAR-10 and CIFAR-100, two more complicated datasets, we use 2000 samples for random projections, 3000 (10000 for CIFAR-100) seed data, and query 5000 samples for 4 iterations. Because Bayesian Coreset usually finds a much smaller batch than requested, for a fair comparison, we let Bayesian Coreset acquire more data than the batch size, and stop the acquisition as long as it has selected a full batch of data. It can be seen in Table 1 and Figure 1 that both SABAL-IHT and SABAL-Greedy show some advantages on Fashion MNIST dataset. On CIFAR-10 and CIFAR-100, we find SABAL-Greedy performs better than SABAL-IHT while outperforming other baselines, including the Bayesian Coreset, one of the current state-of-the-art approach in the literature under Bayesian settings.

Table 1: AUC Score ($\pm$ std.) for different AL methods on Bayesian active learning. AUC measures the overall performance improvement across number of queries. Results show the proposed **SABAL-*** matches or outperforms all baselines.

| Dataset | Bayesian Coreset | SABAL-Greedy | SABAL-IHT | BALD | Random |
|---|---|---|---|---|---|
| Fashion MNIST | $89.53 \pm 0.25$ | $\mathbf{89.83 \pm 0.26}$ | $\mathbf{89.97 \pm 0.23}$ | $89.72 \pm 0.23$ | $88.39 \pm 0.32$ |
| CIFAR10 | $\mathbf{77.73 \pm 0.70}$ | $\mathbf{78.28 \pm 0.53}$ | $77.55 \pm 0.83$ | $77.61 \pm 0.59$ | $76.36 \pm 0.59$ |
| CIFAR100 | $\mathbf{42.40 \pm 0.29}$ | $\mathbf{42.53 \pm 0.34}$ | $42.30 \pm 0.29$ | $41.99 \pm 0.60$ | $42.10 \pm 0.24$ |

Figure 1: Active learning results on Bayesian models. Solid lines and shaded areas represent means and standard deviations of test accuracy over different seeds. Our method especially SABAL-Greedy outperforms most baselines and the SOTA method Bayesian Coreset.

**SABAL for General Active Learning** We then implement our experiments on general convolutional neural networks, including LeNet-5 (LeCun et al., 2015) and VGG-16 (Simonyan & Zisserman, 2014) architectures without any Bayesian layers, using MNIST (LeCun et al., 1998), SVHN (Netzer et al., 2011), and CIFAR-10 (Krizhevsky et al., 2009) datasets. We utilize calibrated prediction of current model with temperature scaling (Guo et al., 2017) to approximate the label distribution $p(\tilde{\boldsymbol{y}}_j \mid \boldsymbol{x}_j, \mathcal{D}_l)$. Because the gradient of the last layer represents the full-model gradient (Ash et al., 2019), we can easily solve the optimization problem with gradient embedding as equation 14. We compare with popular non-Bayesian baselines: Random, Entropy, KCenter, and BADGE.

To evaluate the performance of SABAL, on MNIST dataset with LeNet-5 model, we use 40 seed data, and query 40 samples for 15 iterations. On SVHN and CIFAR-10 with VGG-16 model, which contains more complicated real-world color images, we use 1000 (3000 for CIFAR-10) seed data,

and query 1000 (3000 for CIFAR-10) samples for 5 iterations. The results are shown in Table 2 and Figure 2. In general, SABAL-Greedy also performs better than SABAL-IHT and outperform most baselines, while achieving comparable performance to the strong baseline BADGE, the current SOTA non-Bayesian method in literature, but SABAL requires much less acquisition time than BADGE especially on large models. In addition, most methods perform similarly on CIFAR-10, and we conjecture that for CIFAR-10 each sample is informative enough and thus random selection can achieve good enough performance.

Table 2: AUC Score ($\pm$ std.) for different AL methods on general active learning. AUC measures the overall performance improvement across number of queries. Results show the proposed **SABAL-\*** matches or outperforms all baselines.

| Dataset | BADGE | SABAL-Greedy | SABAL-IHT | KCenter | Entropy | Random |
|---|---|---|---|---|---|---|
| MNIST | $\mathbf{91.24 \pm 0.48}$ | $90.89 \pm 0.38$ | $\mathbf{91.07 \pm 0.45}$ | $89.57 \pm 1.02$ | $90.68 \pm 0.81$ | $86.48 \pm 1.11$ |
| SVHN | $86.92 \pm 0.71$ | $\mathbf{87.23 \pm 0.47}$ | $86.84 \pm 0.57$ | $\mathbf{87.04 \pm 0.80}$ | $86.28 \pm 1.05$ | $85.52 \pm 0.51$ |
| CIFAR10 | $\mathbf{68.20 \pm 0.56}$ | $\mathbf{68.01 \pm 0.66}$ | $67.80 \pm 0.69$ | $67.98 \pm 0.63$ | $67.94 \pm 0.64$ | $67.05 \pm 0.59$ |

Figure 2: Active learning results on general (non-Bayesian) models. Solid lines and shaded areas represent means and standard deviations of test accuracy over different seeds. Our method performs comparable or better than baselines and the SOTA method BADGE.

**Run Time Comparison** Our experiments show that SABAL can achieve comparable performance of SOTA methods while requiring much less acquisition time. We compare the empirical results of runtime complexity of SABAL with other baselines in non-Bayesian active learning experiment as an example. Here, we consider the acquisition time of the first query, where the unlabeled data pool has the largest size compared with later queries. Large models and datasets (SVHN and CIFAR-10 on VGG-16) are used to better illustrate the runtime complexity. Results are shown in Table 3. It can be seen that SABAL requires much less runtime than BADGE especially when queried batch is large, and even less than KCenter in most cases.

Table 3: first query's acquisition time ($\pm$ std.) of different AL methods on two large datasets. **SABAL-\*** shows big runtime advantage.

| Dataset | Method | Time (unit:s) | Dataset | Method | Time (unit:s) |
|---|---|---|---|---|---|
| SVHN | BADGE | $732.18 \pm 26.29$ | CIFAR10 | BADGE | $1207.19 \pm 121.09$ |
| | SABAL-Greedy | $201.65 \pm 3.54$ | | SABAL-Greedy | $333.67 \pm 3.53$ |
| | SABAL-IHT | $211.04 \pm 10.65$ | | SABAL-IHT | $174.28 \pm 3.27$ |
| | KCenter | $309.99 \pm 0.81$ | | KCenter | $258.29 \pm 1.75$ |
| | Entropy | $16.46 \pm 0.29$ | | Entropy | $11.76 \pm 0.03$ |
| | Random | $0.81 \pm 0.02$ | | Random | $1.42 \pm 0.02$ |

## 7 CONCLUSION

We introduce the SABAL as a novel framework that formulates batch active learning as a sparse approximation problem. It balances representation and uncertainty in a principled way, and has the flexibility to adapt to both Bayesian and non-Bayesian models. We realize the SABAL framework as a finite-dimensional optimization problem, efficiently solvable by the proposed greedy or proximal IHT algorithms. Numerical experiments demonstrate the strong performance of SABAL, comparable to the state-of-the-art with lower time complexity. For the future works, although the hyperparameter $\alpha$ offers a controllable trade-off between the variance and bias, it is still not well-understood how to strike the best balance. An in-depth theoretical analysis of the SABAL optimizations, as well as other instantiations of our general framework, would also have the potential to inspire discoveries of even better batch active learning algorithms.

## 8 ETHICS STATEMENT

This work is proposing an active learning approach for more efficient data acquisition and model training, we do not expect any obvious ethical issue from this work.

## 9 REPRODUCIBILITY STATEMENT

We have included the code and instructions to reproduce our work in the supplementary material. We also provide the experiment settings, training details and hyperparameters in the Appendix.

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

# SABAL: Sparse Approximation-based Batch Active Learning

## Appendix

## A  THE OVERALL PROCEDURE

---

**Algorithm 3:** SABAL: Sparse Approximation-based Batch Active Learning

---

**Input:** Intial parameters $\boldsymbol{\theta}$, initial unlabeled pool $\mathcal{D}_u$, initial labeled pool $\mathcal{D}_l = \varnothing$, initial number
   of samples $b_0$, query batch size $b$, number of iterations $T$.

1 Query a random batch $\mathcal{S}_0$ of $b_0$ data from $\mathcal{D}_l$, update $\mathcal{D}_u \leftarrow \mathcal{D}_u \backslash \mathcal{S}_0$ and $\mathcal{D}_l \leftarrow \mathcal{D}_l \cup \mathcal{S}_0$.
2 Train the model using $\mathcal{S}_0$.
3 **for** $t = 1, 2, \ldots, T$ **do**
4   For each data $\boldsymbol{x}_j \in \mathcal{D}_u$, estimate its label distribution $\tilde{\boldsymbol{y}}_j \sim \mathscr{P}(\boldsymbol{x}_j)$.
5   Compute vector $\boldsymbol{g}_j$ by sampling or gradient embedding using 16.
6   Compute $\boldsymbol{v}$, $\Phi$, and $\sigma_j$ for each $\boldsymbol{x}_j \in \mathcal{D}_u$ and form equation 17.
7   Find sparse weight $\boldsymbol{w}$ s.t. $\|\boldsymbol{w}\|_0 = b$ as specified in section 4.
8   Select a batch of data $\mathcal{S}_t = \{\boldsymbol{x}_j \in \mathcal{D}_u \mid \boldsymbol{w}_j > 0\}$ and query their labels.
9   Update $\mathcal{D}_u \leftarrow \mathcal{D}_u \backslash \mathcal{S}_t$ and $\mathcal{D}_l \leftarrow \mathcal{D}_l \cup \mathcal{S}_t$.
10  Reinitialize and retrain the model using updated $\mathcal{D}_l$, update model parameters $\boldsymbol{\theta}$
11 **end**
**Return:** Final model parameters $\boldsymbol{\theta}$.

---

## B  MORE RELATED WORK

Active learning has been widely studied by the machine learning community. As most classic approaches have already been discussed in a detail in Settles (2009); Dasgupta (2011); Hanneke et al. (2014), we will briefly review some recent works in deep active learning.

Existing query strategies can mainly be categorized as uncertainty-based and representation-based. Uncertainty-based approaches look for data samples the model is mostly uncertain about. Meanwhile, under the Bayesian setting, several recent works leverage the Bayesian neural network to well measure the model uncertainty. Gal & Ghahramani (2015; 2016) proves the Monte-Carlo dropout (MC Dropout) as an approximation of performing Bayesian inference, and enables efficient uncertainty estimations in neural networks. Gal et al. (2017) utilizes MC Dropout for approximating posterior distributions and adapts Houlsby et al. (2011) as their uncertainty based acquisition function, and similarly, Kirsch et al. (2019) proposes a batch-mode approach based on Gal et al. (2017) and shows some improvements through a more accurate measurement of mutual information between the data batch and model parameters. While MC Dropout becomes prevalent for uncertainty estimation, Beluch et al. (2018) shows ensemble-based methods lead to better performance because of more calibrated uncertainty estimation, and another recent work Hemmer et al. (2020) also proposes a new uncertainty estimation method by replacing the softmax output of a neural network with the parameter of Dirichlet density. Other non-Bayesian approaches sometimes combine uncertainty estimation with other metrics: Li & Guo (2013) combines an information density measure to maximize the mutual information between selected samples and remaining unlabeled samples under the Gaussian Process setting. Wang et al. (2016) selects data based on several classic uncertainty metrics and incorporate a cost-efficient strategy by pseudo labeling the confident samples.

Representation-based approaches attempt to query diverse data samples that could best represent the overall unlabeled dataset. A recent work proposed by Sener & Savarese (2017) defines the active learning as a core-set selection problem. They derive an upper bound for the core-set loss and construct representative batches by solving a k-Center problem in the feature space. In Geifman & El-Yaniv (2017), the authors also explore the deep active learning with core-sets, but build the core-sets in the farthest-first compression scheme.

## C   PROOFS

**Proposition C.1.** $q : \mathcal{L}(\Theta, \mathbb{R}) \to \mathbb{R}_+$ *defined in ([5](#)) is a shift-invariant seminorm satisfying the following properties:*

1. $q(L_1 + L_2) \le q(L_1) + q(L_2)$ *for* $\forall L_1, L_2 \in \mathcal{L}(\Theta, \mathbb{R})$;                    *(triangle inequality)*

2. $q(cL) = |c|q(L)$ *for* $\forall L \in \mathcal{L}(\Theta, \mathbb{R}), \forall c \in \mathbb{R}$;                    *(absolute homogeneity)*

3. $q(L + c) = q(L)$ *for* $\forall L \in \mathcal{L}(\Theta, \mathbb{R}), \forall c \in \mathbb{R}$;                    *(shift-invariance)*

4. $q(L) = 0$ *if and only if* $L$ *maps every* $\boldsymbol{\theta} \in \Theta$ *to a constant.*

*In other words, $q$ defines a norm in the space of shift-equivalence classes of loss functions.*

*Proof.* Recall that

$$q(L) := \inf_{c \in \mathbb{R}} \|L + c\|_\dagger, \qquad \forall L \in \mathcal{L}(\Theta, \mathbb{R}).$$

We prove the four properties respectively in the following.

1. The triangle inequality is inherited from the sub-additivity of the norm $\|\cdot\|_\dagger$. For $\forall L \in \mathcal{L}(\Theta, \mathbb{R})$, we have

$$\begin{aligned} q(L_1 + L_2) = \inf_{c \in \mathbb{R}} \|L_1 + L_2 + c\|_\dagger &= \inf_{c_1, c_2 \in \mathbb{R}} \|L_1 + L_2 + c_1 + c_2\|_\dagger \\ &\le \inf_{c_1, c_2 \in \mathbb{R}} \|L_1 + c_1\|_\dagger + \|L_2 + c_2\|_\dagger \\ &= (\inf_{c \in \mathbb{R}} \|L_1 + c\|_\dagger) + (\inf_{c \in \mathbb{R}} \|L_2 + c\|_\dagger) \\ &= q(L_1) + q(L_2). \end{aligned}$$

2. The absolute homogeneity is also inherited from the absolute homogeneity of the norm $\|\cdot\|_\dagger$. The case for $c = 0$ is obvious, and for $c \ne 0$ we have

$$\begin{aligned} q(cL) = |c|q(L) = \inf_{c_1 \in \mathbb{R}} \|cL + c_1\|_\dagger &= \inf_{c_1 \in \mathbb{R}} |c| \cdot \|L + c_1/c\|_\dagger \\ &= \inf_{c_2 \in \mathbb{R}} |c| \cdot \|L + c_2\|_\dagger = |c|q(L). \end{aligned}$$

3. By the definition of $q(\cdot)$, we have the shift-invariance of $q(\cdot)$.

4. The "if" part can be proved by definition, *i.e.*, $q(c) = \inf_{c_1 \in \mathbb{R}} \|c_1 + c\|_\dagger = \|0\|_\dagger = 0$.

   For the "only if" part, we need to be more rigorous by defining $f_c$ to be the function that maps $\Theta$ to $c \in \mathbb{R}$. We further define $\mathcal{F} := \{f_c \mid c \in \mathbb{R}\} \subset \mathcal{L}(\Theta, \mathbb{R})$, and we can see $(\mathcal{F}, \|\cdot\|_\dagger)$ is a one-dimensional normed space. Letting $L \in \mathcal{L}(\Theta, \mathbb{R})$ and $q(L) = 0$, we have

$$\inf_{f_{c_\epsilon} \in \mathcal{F}} \|L + f_c\|_\dagger = 0.$$

   Therefore, for $\forall \epsilon > 0, \exists c_\epsilon \in \mathbb{R}$ such that

$$\begin{aligned} \|L + f_{c_\epsilon}\|_\dagger &\le \epsilon \\ \implies \|f_{c_\epsilon}\|_\dagger = \|L + f_{c_\epsilon} - L\|_\dagger &\le \epsilon + \|L\|_\dagger. \end{aligned}$$

   That being said, for $0 < \epsilon < 1$, we have $\|f_{c_\epsilon}\|_\dagger \le 1 + \|L\|_\dagger$. Denote $\mathcal{B} = \{f_c \in \mathcal{F} \mid \|f_c\|_\dagger \le 1 + \|L\|_\dagger\}$, and we can see $\mathcal{B}$ is a closed ball in $\mathcal{F}$. As $\mathcal{F}$ is one-dimensional, by Riesz's lemma we have $\mathcal{B}$ compact.

   As $\lim_{\epsilon \to 0} \|L + f_{c_\epsilon}\|_\dagger = 0$, *i.e.*, $f_{c_\epsilon} \to L$, by the compactness of $\mathcal{B}$ we have $L \in \mathcal{B}$. Therefore, $L$ is also a constant function. Note that this conclusion does not require $\mathcal{L}(\Theta, \mathbb{R})$ to be complete.

$\square$

**Proposition C.2** (Proposition 3.1 Restated). *As $\boldsymbol{w} \in \mathbb{R}_+^{n_u}$ and $\|\boldsymbol{w}\|_0 = b$, by replacing the $\mathscr{P}$ by the improved estimation distribution $\mathscr{P}_{\boldsymbol{w}}$ (equation 9) into (i) in equation 7, we have*

$$\mathbb{E}_{\mathscr{P}_{\boldsymbol{w}}}[q(\tilde{L} - \mathbb{E}_{\mathscr{P}_{\boldsymbol{w}}}[\tilde{L}])] + \mathbb{E}_{\mathscr{P}_{\boldsymbol{w}}}[q(\tilde{L}_{\boldsymbol{w}} - \mathbb{E}_{\mathscr{P}_{\boldsymbol{w}}}[\tilde{L}_{\boldsymbol{w}}])] \leq \sum_{\boldsymbol{x}_j \in \mathcal{D}_u} \mathbf{1}(w_j = 0) \cdot \sigma_j,$$

*where $\sigma_j := \frac{1}{n_u}\mathbb{E}_{\mathscr{P}(\boldsymbol{x}_j)}[q(\ell(\boldsymbol{x}_j, \tilde{\boldsymbol{y}}_j; \cdot) - \mathbb{E}_{\mathscr{P}(\boldsymbol{x}_j)}[\ell(\boldsymbol{x}_j, \tilde{\boldsymbol{y}}_j; \cdot)])]$ is the individual variance, and $\mathbf{1}(\cdot)$ is the indicator function.*

*Proof.* Recall that

$$\tilde{L}_{\boldsymbol{w}}(\boldsymbol{\theta}) := \frac{1}{b}\sum_{\boldsymbol{x}_j \in \mathcal{D}_u} w_j \ell(\boldsymbol{x}_j, \tilde{\boldsymbol{y}}_j; \boldsymbol{\theta}), \qquad\qquad \tilde{L}(\boldsymbol{\theta}) := \frac{1}{n_u}\sum_{\boldsymbol{x}_j \in \mathcal{D}_u} \ell(\boldsymbol{x}_j, \tilde{\boldsymbol{y}}_j; \boldsymbol{\theta}),$$

$$\tilde{\boldsymbol{y}}_j \sim \mathscr{P}_{\boldsymbol{w}}(\boldsymbol{x}_j) := \begin{cases} \mathscr{P}(\boldsymbol{x}_j) & \text{if } w_j = 0 \\ \delta_{\boldsymbol{y}_j^\star} & \text{if } w_j > 0 \end{cases}, \qquad \boldsymbol{w} \in \mathbb{R}_+^{n_u},$$

where $\delta_{\boldsymbol{y}_j^\star}$ denotes the distribution that $\tilde{\boldsymbol{y}}_j$ can only be $\boldsymbol{y}_j^\star$. Therefore, by the definition of $\mathscr{P}_{\boldsymbol{w}}$, we have

$$\mathbb{E}_{\mathscr{P}_{\boldsymbol{w}}(\boldsymbol{x}_j)}\left[q\left(\ell(\boldsymbol{x}_j, \tilde{\boldsymbol{y}}_j; \cdot) - \mathbb{E}_{\mathscr{P}_{\boldsymbol{w}}(\boldsymbol{x}_j)}[\ell(\boldsymbol{x}_j, \tilde{\boldsymbol{y}}_j; \cdot)]\right)\right] = 0, \qquad \text{if } w_j > 0.$$

Plugging the above definitions into $\mathbb{E}_{\mathscr{P}_{\boldsymbol{w}}}[q(\tilde{L}_{\boldsymbol{w}} - \mathbb{E}_{\mathscr{P}_{\boldsymbol{w}}}[\tilde{L}_{\boldsymbol{w}}])]$ we have

$$\mathbb{E}_{\mathscr{P}_{\boldsymbol{w}}}[q(\tilde{L}_{\boldsymbol{w}} - \mathbb{E}_{\mathscr{P}_{\boldsymbol{w}}}[\tilde{L}_{\boldsymbol{w}}])] = \mathbb{E}_{\mathscr{P}_{\boldsymbol{w}}}\left[q\left(\frac{1}{n_u}\sum_{\boldsymbol{x}_j \in \mathcal{D}_u} w_j\left(\ell(\boldsymbol{x}_j, \tilde{\boldsymbol{y}}_j; \cdot) - \mathbb{E}_{\mathscr{P}_{\boldsymbol{w}}(\boldsymbol{x}_j)}[\ell(\boldsymbol{x}_j, \tilde{\boldsymbol{y}}_j; \cdot)]\right)\right)\right]$$

$$= \mathbb{E}_{\mathscr{P}_{\boldsymbol{w}}}\left[q\left(\frac{1}{n_u}\sum_{\boldsymbol{x}_j \in \mathcal{D}_u} \mathbf{1}(w_j > 0)w_j\left(\ell(\boldsymbol{x}_j, \tilde{\boldsymbol{y}}_j; \cdot) - \mathbb{E}_{\mathscr{P}_{\boldsymbol{w}}(\boldsymbol{x}_j)}[\ell(\boldsymbol{x}_j, \tilde{\boldsymbol{y}}_j; \cdot)]\right)\right)\right] = 0. \tag{18}$$

Therefore, we only need to care about the $\mathbb{E}_{\mathscr{P}_{\boldsymbol{w}}}[q(\tilde{L} - \mathbb{E}_{\mathscr{P}_{\boldsymbol{w}}}[\tilde{L}])]$.

$$\mathbb{E}_{\mathscr{P}_{\boldsymbol{w}}}[q(\tilde{L} - \mathbb{E}_{\mathscr{P}_{\boldsymbol{w}}}[\tilde{L}])] = \mathbb{E}_{\mathscr{P}_{\boldsymbol{w}}}\left[q\left(\frac{1}{n_u}\sum_{\boldsymbol{x}_j \in \mathcal{D}_u} \ell(\boldsymbol{x}_j, \tilde{\boldsymbol{y}}_j; \cdot) - \mathbb{E}_{\mathscr{P}_{\boldsymbol{w}}(\boldsymbol{x}_j)}[\ell(\boldsymbol{x}_j, \tilde{\boldsymbol{y}}_j; \cdot)]\right)\right]$$

$$\leq \sum_{\boldsymbol{x}_j \in \mathcal{D}_u} \frac{1}{n_u}\mathbb{E}_{\mathscr{P}_{\boldsymbol{w}}(\boldsymbol{x}_j)}\left[q\left(\ell(\boldsymbol{x}_j, \tilde{\boldsymbol{y}}_j; \cdot) - \mathbb{E}_{\mathscr{P}_{\boldsymbol{w}}(\boldsymbol{x}_j)}[\ell(\boldsymbol{x}_j, \tilde{\boldsymbol{y}}_j; \cdot)]\right)\right]$$

$$= \sum_{\boldsymbol{x}_j \in \mathcal{D}_u} \mathbf{1}(w_j = 0)\frac{1}{n_u}\mathbb{E}_{\mathscr{P}}\left[q\left(\ell(\boldsymbol{x}_j, \tilde{\boldsymbol{y}}_j; \cdot) - \mathbb{E}_{\mathscr{P}(\boldsymbol{x}_j)}[\ell(\boldsymbol{x}_j, \tilde{\boldsymbol{y}}_j; \cdot)]\right)\right]$$

$$= \sum_{\boldsymbol{x}_j \in \mathcal{D}_u} \mathbf{1}(w_j = 0) \cdot \sigma_j, \tag{19}$$

where the inequality is by the triangle inequality and the absolute homogeneity of $q(\cdot)$ (Proposition C.1). Combining equation 18 and equation 19, we have the proposition proved. $\square$

## D   OMITTED ALGORITHMS

In this section we present the two sub-procedures, *i.e.*, line search and de-bias, shared by two main optimization algorithms (Algorithm 1&2), as well as how the optimization (line 6) in Algorithm 2 is solved optimally.

The line search sub-procedure (Algorithm 4) optimally solve the problem of $\arg\min_{\mu \in \mathbb{R}} f_1(\boldsymbol{w} - \mu\boldsymbol{u})$, *i.e.*, given a direction $\boldsymbol{u}$ what is the best step size to move the $\boldsymbol{w}$ along $\boldsymbol{u}$. The de-bias sub-procedure (Algorithm 5) adjusts a sparse $\boldsymbol{w}$ in its own sparse support for a better solution.

---

**Algorithm 4:** LineSearch($\boldsymbol{u}, \boldsymbol{w}$)

**Input:** direction $\boldsymbol{u}$; starting point $\boldsymbol{w}$.
**Output:** step size $\mu$.

1   $\mu \leftarrow \frac{\langle \Phi \boldsymbol{w} - \boldsymbol{v}, \Phi \boldsymbol{u} \rangle + \beta \langle \boldsymbol{w} - \boldsymbol{1}, \boldsymbol{u} \rangle}{\|\Phi \boldsymbol{u}\|_2^2 + \beta \|\boldsymbol{u}\|_2^2}$    *(optimal $\mu$)*

**Return:** $\mu$

---

**Algorithm 5:** De-bias($\boldsymbol{w}$)

**Input:** starting point $\boldsymbol{w}$.
**Output:** improved $\boldsymbol{w}$.

1   $\boldsymbol{u} \leftarrow [\nabla f_1(\boldsymbol{w})]_{\text{supp}(\boldsymbol{w})}$   *(in-support grad.)*
2   $\mu \leftarrow \text{LineSearch}(\boldsymbol{u}, \boldsymbol{w})$
3   $\boldsymbol{w} \leftarrow \boldsymbol{w} - \mu \boldsymbol{u}$    *(in-support adjustment)*

**Return:** $\boldsymbol{w}$

---

Recall the inner optimization (line 6) of Algorithm 2 is

$$\boldsymbol{w} \leftarrow \argmin_{\boldsymbol{w} \in \mathbb{R}_+^{n_u}, \|\boldsymbol{w}\|_0 \leq b} \frac{1}{2}\|\boldsymbol{w} - \boldsymbol{s}\|_2^2 + f_2(\boldsymbol{w}).$$

Noting that $\frac{1}{2}\|\boldsymbol{w} - \boldsymbol{s}\|_2^2 + f_2(\boldsymbol{w}) = \sum_{j \in [n_u]}(\frac{1}{2}(w_j - s_j)^2 - \alpha \sigma_j^2)$, this step can be done optimally by simply picking the top-$b$ elements, as shown in the following. Given a $b$-sparse support set $\mathcal{S} \subset [n_u]$, we can see that

$$\min_{\boldsymbol{w} \in \mathbb{R}_+^{n_u}, \text{supp}(\boldsymbol{w}) \subseteq \mathcal{S}} \sum_{j \in [n_u]}(\tfrac{1}{2}(w_j - s_j)^2 - \alpha \sigma_j^2) = \sum_{j \in \mathcal{S}}(\tfrac{1}{2}[-s_j]_+^2 - \alpha \sigma_j^2).$$

Therefore, line 6 in Algorithm 2 can be done by: (1) find the $b$ smallest $(\frac{1}{2}[-s_j]_+^2 - \alpha \sigma_j^2)$, denoting the resulting $b$-sparse index set as $\mathcal{S}^\star$; (2) let $\boldsymbol{w} \leftarrow [[\boldsymbol{s}]_{\mathcal{S}^\star}]_+$.

## E   More Experiment Results

### E.1   Ablation Study: trade-off of uncertainty and representation

We perform an ablation study to understand better the trade-off between the variance and the bias terms in our final formulation equation 15. To remove the bias term, we query the data with top variances. To remove the variance term, we query the data by only minimizing the approximation bias, i.e., setting $\alpha = 0$, under both IHT and Greedy optimizations respectively. We take two datasets MNIST and CIFAR-10 in the Bayesian experiment as examples. Results in Figure 3 demonstrate the necessity of taking both uncertainty and representation into consideration during the data acquisitions for ideal performance, while for some datasets like CIFAR-10, the variance contributes much more significantly.

## F   Implementation Details

All experiments are written in PyTorch 1.8.1. All hyper-parameters are chosen to ensure models achieve good and stable performance on each dataset, and they are kept identical for all active learning approaches.

Figure 3: Ablation Study Results on Bayesian models.

### F.1   Bayesian Active Learning Experiment

**Model Architecture**   We use the exact same model as (Pinsler et al., 2019), it is a Bayesian neural network consisting of a ResNet-18 (He et al., 2016) feature extractor followed by a fully connected layer with a ReLU activation, and a final layer allows sampling by local reparametrization (Kingma et al., 2015) with a softmax activation.

**Optimization and Hyperparameter Selection**   Due to larger models and more complicated classification tasks, e.g., CIFAR-100, data augmentation(including random cropping and random horizontal flipping) and learning rate scheduler are used in this experiment to achieve good model performance. The model is optimized with the Adam (Kingma & Ba, 2014) optimizer using default exponential decay rates (0.9, 0.999) for the moment estimates. Table 4 shows the hyper-parameters in experiment

on Bayesian batch active learning, where $bs$ denotes the batch size in dataloader during the model training, $lr$ denotes the learning rate, and $wd$ denotes the weight decay. The hyper-parameters are chosen through grid search.

Table 4: Hyperparameters used in Bayesian active learning experiment

| Dataset | Method | Epoch | $bs$ | $\alpha$ | $\beta$ | $lr$ | $wd$ |
|---------|--------|-------|------|----------|---------|------|------|
| Fashion MNIST | SABAL-IHT | 200 | 256 | 1 | $10^{-3}$ | 0.001 | $5 \times 10^{-4}$ |
| Fashion MNIST | SABAL-Greedy | 200 | 256 | 2 | 0.5 | 0.001 | $5 \times 10^{-4}$ |
| CIFAR-10 | SABAL-IHT | 200 | 256 | 1 | $10^{-6}$ | 0.001 | $5 \times 10^{-4}$ |
| CIFAR-10 | SABAL-Greedy | 200 | 256 | 2 | 1 | 0.001 | $5 \times 10^{-4}$ |
| CIFAR-100 | SABAL-IHT | 200 | 256 | 1 | $10^{-6}$ | 0.001 | $5 \times 10^{-4}$ |
| CIFAR-100 | SABAL-Greedy | 200 | 256 | 1 | 0.5 | 0.001 | $5 \times 10^{-4}$ |

### F.2 GENERAL ACTIVE LEARNING EXPERIMENT

**Model Architecture** On MNIST dataset, we use LeNet-5 model (LeCun et al., 2015). On SVHN and CIFAR10 datasets, we use VGG-16 model (Simonyan & Zisserman, 2014).

**Optimization and Hyperparameter Selection** All models are trained using the cross entropy loss with SGD optimizer, and no data augmentation or learning rate scheduler is used. Tabel 5 shows the hyper-parameters in experiment on general batch active learning, where $bs$ denotes the batch size in dataloader during the model training, $lr$ denotes the learning rate, $m$ denotes the momentum, and $wd$ denotes the weight decay. The hyper-parameters are chosen through grid search.

Table 5: Hyperparameters used in general active learning experiment

| Dataset | Method | Epoch | $bs$ | $\alpha$ | $\beta$ | $lr$ | | $wd$ |
|---------|--------|-------|------|----------|---------|------|------|------|
| MNIST | SABAL-IHT | 150 | 32 | $10^{-8}$ | $10^{-4}$ | 0.01 | 0.9 | $5 \times 10^{-4}$ |
| MNIST | SABAL-Greedy | 150 | 32 | $10^{-8}$ | $10^{-1}$ | 0.01 | 0.9 | $5 \times 10^{-4}$ |
| SVHN | SABAL-IHT | 150 | 128 | $10^{-8}$ | $10^{-4}$ | 0.01 | 0.9 | $5 \times 10^{-4}$ |
| SVHN | SABAL-Greedy | 150 | 128 | $10^{-1}$ | $10^{-1}$ | 0.01 | 0.9 | $5 \times 10^{-4}$ |
| CIFAR-10 | SABAL-IHT | 100 | 128 | $10^{-8}$ | $10^{-6}$ | 0.001 | 0.9 | $5 \times 10^{-4}$ |
| CIFAR-10 | SABAL-Greedy | 100 | 128 | $10^{-2}$ | $10^{-1}$ | 0.001 | 0.9 | $5 \times 10^{-4}$ |

