# OpenReview forum: "SABAL: Sparse Approximation-based Batch Active Learning"
_ICLR.cc/2022/Conference — ICLR 2022 Submitted_

### Official Review · Reviewer_ECiF · 2021-10-19

**Correctness:** 3
**Technical Novelty And Significance:** 3
**Empirical Novelty And Significance:** 2
**Recommendation:** 5
**Confidence:** 4

**Main Review:**

Strengths:
* The formulation of batch active learning as sparse approximation is interesting and seems novel.
* The paper considers two choices for the norm for Bayesian and non-Bayesian settings.


Weaknesses:
* The formulation between batch active learning and sparse approximation requires a series of approximations.
* In short, there are multiple approximation techniques used in this paper while the empirical gain is minimal. Random approach performs extremely well for batch active learning task without relying on complex computation.
* Missing a key comparison with Batch BALD. Note that the performance of BALD with $b>1$ is different from BatchBALD with $b>1$ as showed in [1].

Comments/Questions:
* Since the original objective function is intractable. The paper proposes to minimize the upper bound. However, the approximation relies on the tightness of the upper bound. It is questionable about the gap in Eq. (7).

* Then the final approximation form is nonconvex sparsity constraint and discontinuous objective function. To solve this approximation, the paper utilizes two approximation algorithms including the greedy and the iterative hard thresholding to optimize the objective function.

* In Eq (11), the variance term (right handside) will tend to enforce $w_j=0$ as much as possible. The effect is similar in using L0 regularisation.
* where $\alpha$ in Eq. (15) comes from?


Suggestions:
* Ablation study with respect to the batch size b
* while the reviewer has seen the ablation study of trading-off uncertainty and representation (E.1), the reviewer thinks it may be useful to empirically study the performance w.r.t. different choice of $\alpha$.

Presentation:
* The paper is in general well written and easy to follow.

Reproducibility:
* The results appear to be reproducible. The code is included in the appendix.

[1] Kirsch, A., Van Amersfoort, J. and Gal, Y., 2019. Batchbald: Efficient and diverse batch acquisition for deep bayesian active learning. Advances in neural information processing systems, 32, pp.7026-7037.



**Summary Of The Paper:**

The paper proposes SABAL as a framework to formulate batch active learning as a sparse approximation problem. The paper considers the SABAL framework as a finite-dimensional optimization problem, efficiently solvable by the proposed greedy or proximal IHT algorithms.

Numerical experiments demonstrate the strong performance of SABAL, comparable to the state-of-the-art with lower time complexity.

Since directly optimising the original objective function is intractable, SABAL proposes to optimise (minimise) the upper bound which is obtained by the triangle inequality including the (i) variance and (ii) approximation bias.


**Summary Of The Review:**

Active learning is helpful to acquire labelled data for supervised learning. Formulating the batch active learning as sparse approximation problem is interesting.

However, the reviewer has three main concerns: (i) while the maths looks correct, there are a lot of approximations used in finding the solution. These approximations may affect the overall performance and hinder the applicability of the paper. (ii) the empirical performance is minimal comparing to the random baseline. (iii) missing the key comparison to Batch BALD [1].

---

> ### Author Response · Authors · 2021-11-22
> **Response to Reviewer ECiF**
>
> We would like to thank the reviewer for the valuable feedback and comments. We are glad that you find this work interesting. We address your concerns in detail below.
>
> $\textbf{(1)}$ Regarding “there’re a lot of approximations which might affect the empirical performance.”
>
> Thanks for the good suggestion. Some kinds of approximations are still inevitable, for example in our paper Section 3 when measuring the difference between two functions is intractable, we have to formulate such a problem as finite-dimensional optimization. It is indeed interesting to understand if the approximations could be further quantified -- this is also the problem that we would like to investigate in future work. However, more precise bounds are certainly non-trivial as far as we can see. We believe that the key contribution of this work is being the first to propose a novel general framework for the batch active learning problem from a functional approximation perspective. We believe our approach has the potential to solve active learning problems for different settings and applications.
>
> $\textbf{(2)}$ Regarding “the improvement of empirical performance is marginal.”
>
> Our main target is to propose a novel general framework for batch active learning, and our methods run at a lower runtime compared to these SOTA methods that reach comparable performance.
>
> $\textbf{(3)}$ Regarding the comparison to BatchBALD [1].
>
> Bayesian Coreset [2] is one of the most recent SOTA methods in Bayesian active learning that we mainly focus on, and this work doesn’t compare their performance with BatchBALD. But we agree with reviewers that BatchBALD is different from BALD and we will take BatchBALD as an additional baseline in the final version.
>
> $\textbf{(4)}$ Regarding the $\alpha$ in Equation 15.
>
> $\alpha$ as a hyper-parameter can offer a trade-off between bias and variance.
>
> $\textbf{Reference}$
>
> [1] Kirsch, A., Van Amersfoort, J. and Gal, Y., 2019. Batchbald: Efficient and diverse batch acquisition for deep bayesian active learning. Advances in neural information processing systems, 32, pp.7026-7037.
>
> [2] Pinsler et al. Bayesian batch active learning as sparse subset approximation. NeurIPS 2019.

---

> > ### Comment · Reviewer_ECiF · 2021-11-29
> > **Thank you for your response.**
> >
> > Thanks for providing the detailed response. I would like to acknowledge that I have read the response.
> > By reading other reviews, I can see that the paper is unfortunately still under the acceptance bar. I hope the authors can take the reviewer's feedbacks in a constructive way to improve the paper for future submissions.

---

### Official Review · Reviewer_dVWY · 2021-11-01

**Correctness:** 3
**Technical Novelty And Significance:** 3
**Empirical Novelty And Significance:** 2
**Recommendation:** 5
**Confidence:** 4

**Main Review:**

The re-formulation of the active learning problem into a sparse approximation problem is interesting (which I haven't seen elsewhere before). The empirical results also show advantages of the proposed method (under certain approximations) over existing ones. The paper is generally well-written. I summarize my concerns as follows.

1. Is it true that the authors implicitly assume that they are studying the problem in a realizable case where there is no labeling noise in the Y domain?
2. The sparse approximation problem aims to optimize over $w \in {\mathbb R}^{n_u}_{+}$. However, it seems to me that it's more like optimizing over $w \in \{0, 1\}^{n_u}$? Otherwise, Eq (9) doesn't really make sense to me, e.g., what does it really mean to have $w_j = 0.01$?
3. The proposed method focuses on the batched active learning setting, what would happen in the case of sequential labeling?

**Summary Of The Paper:**

This paper formulate the batched active learning problem as a sparse approximation problem. The authors also provide algorithms to solve the sparse approximation problem. Empirical evaluations show the advantages of the proposed methods.

**Summary Of The Review:**

Based on the main review above, I vote for a weak rejection of the current version.

---

> ### Author Response · Authors · 2021-11-22
> **Response to Reviewer dVWY**
>
> We would like to thank the reviewer for the valuable comments and constructive suggestions. We hope the following clarifications to your questions could resolve your concerns.
>
> $\textbf{(1)}$ Regarding whether "the author assumes there’s no noise in the Y domain?"
>
> In Active Learning, we cannot access the labels before the data has been queried, so we can only approximate the label as a distribution and take the expectation. The bias caused by this approximation in unlabeled data is one of the fundamental problems in AL. We expect that a study of the theoretical aspects of our framework will be an intriguing future work -- particularly given that most of the non-trivial active learning methods do not have in-depth theoretical studies.
>
> $\textbf{(2)}$ Regarding whether "the sparse approximation seems like optimizing over $w\in\{0,1\}$?"
>
> The sparse weight $w$ is actually real-valued instead of binary. The two optimization algorithms proposed in our paper, i.e., SABAL-Greedy and SABAL-IHT will both enforce the weight $w$ to be sparse by keeping only its top $b$ entries as the final optimization result. If the optimization algorithm gives $w_j = 0.01$ in its intermediate step as an example the reviewer mentioned, $w_j$ may be zeroed out by the sparse selection step in the output $w$ and $x_j$ will not be queried in this iteration.
>
> $\textbf{(3)}$ “The proposed method is batch active learning, what about sequential labeling?”
>
> As mentioned in the introduction section of our paper, in the case of sequential labeling, every time we have queried one data point, we have to retrain the whole model again. The number of re-trainings equals the number of data queried in total. Both the computation and time for frequent model retraining are astronomical, so sequential labeling is unfeasible in practical active learning, especially for deep models.

---

### Official Review · Reviewer_3Nq2 · 2021-11-01

**Correctness:** 4
**Technical Novelty And Significance:** 4
**Empirical Novelty And Significance:** 3
**Recommendation:** 5
**Confidence:** 4

**Main Review:**

Strengths: This work decompose the upper bound of the sparse approximation problem as bias and variance terms, which naturally provides the uncertainty and representativeness during AL processes. By transforming the sparse approximation problem to finite-dim optimization problem, and solve w to implement subset selection of AL. The idea is interesting and novel.

Weaknesses: my questions focused on the experimental part.

1) The quota of different experiments are inconsistent, i.e., in Figure 1, the quota on CIFAR10 dataset is 18000, while in Figure 2, the quota of CIFAR10 is 23000. The change of quota would influence the value of AUC score.
2) For the chosen of baseline, why not employ BatchBALD[1] instead of BALD? SABAL should compare with hybrid AL sampling strategies, BatchBALD would be more suitable for comparison.
3) From the experimental results, although SABAL reduce the acquisition time, the improvement of performance is marginal. However, there is no shortage of AL algorithms that perform better and run faster, e.g., WAAL [2]. WAAL achieved significant performance improvement, while the acquisition time cost is low (see Table 1 in [2]).

References:

[1] Kirsch A, Van Amersfoort J, Gal Y. Batchbald: Efficient and diverse batch acquisition for deep bayesian active learning[J]. Advances in neural information processing systems, 2019, 32: 7026-7037.

[2] Shui C, Zhou F, Gagné C, et al. Deep active learning: Unified and principled method for query and training[C]//International Conference on Artificial Intelligence and Statistics. PMLR, 2020: 1308-1318.

**Summary Of The Paper:**

This work formulated batch active learning as sparse approximation problem and provided bayesian & non-bayesian versions of their framework. Experimental results shows their effectiveness of reducing the acquisition time, especially when compare with other hybrid AL sampling strategies, such as BADGE.

**Summary Of The Review:**

This paper provides a novel perspective of combining uncertainty and representativeness criteria together in deep active learning. However, the performance gain compared with baseline models is marginal.

---

> ### Author Response · Authors · 2021-11-22
> **Response to Reviewer 3Nq2**
>
> We would like to thank the reviewer for the careful and valuable comments. We address your concerns and questions in the following.
>
> $\textbf{(1)}$ Regarding “the quota of different experiments are different.”
>
> The quota is different because they are trained with different models and datasets, which means their performance is not comparable, so we pick a reasonable quota to let each model achieve reasonable performance (convergence). For instance, a larger model usually requires more data for training. As long as the evaluation uses the same model and the same dataset, we believe the comparison of different methods is fair. This is also a common practice in the active learning literature.
>
> $\textbf{(2)}$ Regarding comparing with Batch-BALD [1] instead of BALD.
>
> Bayesian Coreset [2] is one of the most recent SOTA methods in Bayesian active learning that we mainly focus on, and this work doesn’t compare their performance with BatchBALD. But we agree with reviewers that BatchBALD is different from BALD and we will take BatchBALD as an additional baseline in the final version.
>
> $\textbf{(3)}$ “The improvement of the performance of SABAL is marginal, there are other AL approaches such as WAAL [3] which is better and faster.”
>
> Our main target is to propose a novel general framework for batch active learning, and our methods run at a lower runtime compared to these SOTA methods that reach comparable performance, but we won't deny that WAAL is another SOTA work with strong empirical performance in general active learning settings.
>
> $\textbf{Reference}$
>
> [1] Kirsch, A., Van Amersfoort, J. and Gal, Y., 2019. Batchbald: Efficient and diverse batch acquisition for deep bayesian active learning. Advances in neural information processing systems, 32, pp.7026-7037.
>
> [2] Pinsler et al. Bayesian batch active learning as sparse subset approximation. NeurIPS 2019.
>
> [3] Shui C, Zhou F, Gagné C, et al. Deep active learning: Unified and principled method for query and training, International Conference on Artificial Intelligence and Statistics. PMLR, 2020: 1308-1318.

---

### Official Review · Reviewer_DQaH · 2021-11-01

**Correctness:** 4
**Technical Novelty And Significance:** 3
**Empirical Novelty And Significance:** 2
**Recommendation:** 5
**Confidence:** 4

**Main Review:**

Although the algorithm features favorable empirical results and a nice conceptual framework, I have some concerns about the details of the algorithm algorithm and evaluation.

How are \alpha, \beta, and \tau chosen? Are they optimized for each dataset? If so, this may be a major drawback of the proposed method.

This optimization seems to have many heuristics and approximations: the choice of norm for the Bayesian and non-Bayesian settings, the L2 regularization, "squaring all of the terms for the ease of optimization", the "de-biasing" step, and using only the neural network's last layer. It would be very helpful if the effect of these approximations could be bounded or quantified. I'm concerned about the scenario that heuristics were added to the algorithm until the algorithm worked well, rather than being theoretically motivated.

If only using the neural network's last layer, isn't the loss convex, and it'd be better to match the loss optimizer rather than the loss (which is indirect), at least in the non-Bayesian setting?

Why are the dataset for the Bayesian setting (Fashion MNIST, CIFAR 10, CIFAR 100) different from the non-Bayesian setting (MNIST, SVHN, CIFAR 10)? Is it possible that all methods could be run on the 5 datasets presented?

**Summary Of The Paper:**

This paper provides a framework which attempts to find a weighted subset of the unlabeled pool that matches the loss on all the points. This paper introduces a seminorm on the loss functions and proposes an optimization method to minimize this. Somewhat favorable results are shown for vision datasets.

**Summary Of The Review:**

Without the above concerns addressed, this paper could have major drawbacks. Thus, for the time being, I'm recommending rejection.

---

> ### Author Response · Authors · 2021-11-22
> **Response to Reviewer DQaH**
>
> We would like to thank the reviewer for the valuable comments and constructive suggestions. We hope the following clarifications to your questions could resolve your concerns.
>
> $(\textbf{1})$ Regarding “How are hyperparameters $\alpha$, $\beta$, $\tau$ chosen?”
>
> They are hyper-parameters to be tuned like other hyper-parameters. Automatically finding the optimal balance between bias and variance in active learning is non-trivial and challenging, which is an interesting future work.
>
> $(\textbf{2})$ Regarding “The optimization has many heuristics and approximations.”
>
> Some kinds of approximations are still inevitable, for example in our paper Section 3 when measuring the difference between two functions is intractable, we have to formulate such a problem as finite-dimensional optimization. It is indeed interesting to understand if the approximations could be further quantified -- this is also the problem that we would like to investigate in future work. However, more precise bounds are certainly non-trivial as far as we can see. We believe that the key contribution of this work is being the first to propose a novel general framework for the batch active learning problem from a functional approximation perspective. We believe our approach has the potential to solve active learning problems for different settings and applications.
>
> $(\textbf{3})$ “If only using the last layer of the neural net, it’s better to match the loss optimizer.”
>
> In our experiment of non-Bayesian active learning, we do use the gradient of the last layer to represent the full-model gradient, following what the BADGE paper [1] suggests. Otherwise, using the full-model gradient would be too computationally expensive, especially for deep models. For “match the loss optimizer”, it might be not clear what the reviewer means, e.g., what does the “loss optimizer” here represent?
>
> $(\textbf{4})$ “Why is the dataset for Bayesian different from Non-Bayesian setting?”
>
> There isn’t a particular reason to choose these datasets, and we select some popular datasets that current literature uses for comparisons. For the Bayesian setting, because Bayesian-Corset [2] is the SOTA method that we mainly compare with, we used the same datasets they reported in the original paper, except we add CIFAR100 dataset since it is a hard classification task and interesting to compare. We think three different datasets, from small to large scales, can already demonstrate the overall performance of the proposed active learning method. If time permits, we will consider running experiments on more datasets.
>
> $\textbf{References}$
>
> [1] Ash et al. Deep batch active learning by diverse, uncertain gradient lower bound. ICLR 2020.
>
> [2] Pinsler et al. Bayesian batch active learning as sparse subset approximation. NeurIPS 2019.

---

> > ### Comment · Reviewer_DQaH · 2021-11-22
> > **Reply to response**
> >
> > Thank you for your response.
> >
> > For (3), to be more specific, I mean the "loss minimizer" or the parameters of the last layer that minimize the loss. For neural networks, I see an advantage to trying to match the loss function of the population with the loss function of the actively learned subset, because the loss in neural network parameter optimization is nonconvex. However, in the convex case, there is a single global minimizer. Why not devise a strategy so that the active learning subset's loss minimizer is close to the population loss minimizer? It seems to me that matching the loss, then minimizing the loss and hoping the minimizer's are close is indirect.

---

> > > ### Author Response · Authors · 2021-11-23
> > > **Response**
> > >
> > > Thanks to the reviewer’s clarification, the reviewer suggests instead of matching the loss function, it might be better to directly match the optimal parameter (weights) of the last layer that can minimize the loss function because in this case, the loss function is convex. However, this might not be realizable in practice. Finding the optimal weights of the last layer to minimize the full set loss may be trivial, but selecting a subset of data to match the optimal weights can be infeasible since we need to enumerate all possible subsets. Our current method, by computing the gradient of loss function w.r.t the last layer, can be viewed as an approximation of this process. From the perspective of optimizing the neural network, since first-order methods are commonly used to update the parameter, as long as the gradient of loss function w.r.t to the parameter can be matched, the optimization results should be also close to each other, which implies what the reviewer suggests. Moreover, as the reviewer also points out, the idea of ‘matching minimizers’ may only be applied to convex functions, which limits its potential on non-convex objectives. However, we agree with the reviewer that this is an interesting idea, especially if the enumeration can be avoided.

---

### Decision · Program_Chairs · 2022-01-20

**Decision:**

Reject

**Comment:**

This paper presents a batch active learning approach (where in each active learning round, instead of a single input, we wish to select several inputs to be labeled). The paper attempts to solve this problem by posing it as a sparse approximation problem and shows that their approach performs favorably as compared to some of the existing methods such as BALD and Bayesian Coresets for batch active learning.

While the reviewers appreciated the basic idea and the general framework, there were several concerns from the reviewers (as well as myself upon reading the manuscript). Firstly, the idea of batch active learning as a sparse subset selection problem is not new (Pinsler et al, 2019). While previous methods such as (Pinsler et al, 2019) have used ideas such as Coresets, this paper uses sparse optimization techniques such as Greedy and IHT. Moreover, there were concerns about experimental settings relying on various heuristics, and lack of a more extensive and thorough comparison with important baselines, such as BatchBALD and others, which the authors acknowledged.

The reviewers have read the authors' response and engaged in discussion but their assessment remained unchanged. Based on their assessment and my own reading of the manuscript, the paper does not seem to be ready for publication. The authors are advised to consider the points raised in the reviews which I hope will help strengthen the paper for a future submission.